# ONE STEP FURTHER WITH MONTE-CARLO SAMPLER TO GUIDE DIFFUSION BETTER

**Minsi Ren, Wenhao Deng, Ruiqi Feng, Tailin Wu** *
AI for Scientific Simulation and Discovery Lab, Westlake University
{renminsi, wutailin}@westlake.edu.cn

## ABSTRACT

Stochastic differential equation (SDE)-based generative models have achieved substantial progress in conditional generation via **training-free** differentiable loss-guided approaches. However, existing methodologies utilizing posterior sampling typically confront a substantial estimation error, which results in inaccurate gradients for guidance and leading to inconsistent generation results. To mitigate this issue, we propose that performing an **a**dditional **b**ackward denoising step and **M**onte-Carlo **s**ampling (ABMS) can achieve better guided diffusion, which is a plug-and-play adjustment strategy. To verify the effectiveness of our method, we provide theoretical analysis and propose the adoption of a *dual-focus evaluation framework*, which further serves to highlight the critical problem of *cross-condition interference* prevalent in existing approaches. We conduct experiments across various task settings and data types, mainly including conditional online handwritten trajectory generation, image inverse problems (inpainting, super resolution and gaussian deblurring) molecular inverse design and so on. Experimental results demonstrate that our approach can be effectively used with higher order samplers and consistently improves the quality of generation samples across all the different scenarios. Our code is available at https://github.com/AI4Science-WestlakeU/ABMS.

## 1 INTRODUCTION

In recent years, with the rapid advancement of diffusion model techniques (Sohl-Dickstein et al., 2015; Ho et al., 2020; Song et al., 2021b; Song & Ermon, 2019), conditional diffusion models have emerged as a powerful paradigm in generative AI for solving generative tasks such as high-resolution image synthesis, image restoration, text-to-image generation, 3D shape modeling, molecular design and so on (Dhariwal & Nichol, 2021; Rombach et al., 2022; Zhang et al., 2023; Mardani et al., 2023). By incorporating conditional information (*e.g.*, text descriptions, class labels, degraded images, specific molecular property) into the diffusion process, these models enable fine-grained control over generated outputs. However, despite significant attention, conditional generation still faces a series of challenges in terms of cost and generalizability: typical conditional generation approaches such as classifier guidance (Dhariwal & Nichol, 2021) and classifier-free guidance (Ho & Salimans, 2022) either require additional task-specific training of diffusion models or necessitate training extra noise-compatible conditional discriminators, thus limiting the range of applications. Moreover, for inverse problems that require highly precise conditions, such as the quantum numerical attributes of molecules, the classifier-free paradigm is not applicable.

In this context, a highly versatile approach is the *training-free* guidance methods (Wallace et al., 2023; Chung et al., 2022; Lugmayr et al., 2022; Dou & Song, 2024; Cardoso et al., 2024; Feng et al., 2023) for conditional generation, using off-the-shelf loss guidance during denoising to achieve satisfactory results. These approaches avoid task-specific training or extra conditional networks by directly leveraging pre-defined loss signals, enabling plug-and-play conditional generation across diverse scenarios. Among these methods, DPS (Chung et al., 2023) is a prominent approach leveraging diffusion models to tackle inverse problems, with a series of subsequent improvements. Recent studies (He et al., 2024; Yang et al., 2024) have made some improvements by enforcing manifold

---

*Corresponding author

preservation during the guidance process, thereby enabling the use of larger guidance step sizes in denoising. This advancement not only accelerates sampling efficiency but also enhances the stability of conditional generation, fostering a stronger alignment between theoretical foundations and practical utility.

Nevertheless, despite recent progress, we observe that the majority of existing methods using the gradient direction delivered by the plain DPS formulation as guidance, suffer from systematically biased gradients. Empirically, *guiding the sample toward one condition frequently perturbs other conditions that are intended to remain decoupled*, revealing an inherent cross-talk implicit in the gradient estimate. Motivated by this observation, we argue that the evaluation of a guidance method must simultaneously account for two facets: (i) the degree of alignment between the generated samples and the target condition, and (ii) the preservation of global properties, *e.g.*, FID for images or molecular stability in drug design. Experiments exhibit the trade-off in practice: as the guidance weight increases, compliance with the specified condition improves at the systematic expense of sample quality, manifested by rising FID or diminished molecular stability. Consequently, reporting only alignment metrics paints an incomplete picture and risks selecting operating points that violate downstream requirements.

Furthermore, we analyze the root cause of the problems in existing methods: the large bias in estimating the conditional expectation. To address this issue, we propose ABMS, a strategy that effectively controls estimation bias through Monte Carlo sampling, yielding more accurate guidance gradients. We demonstrate its effectiveness across multiple tasks and data types. Our contributions are summarized as follows:

- We highlight the limitations of previous methods: the significant estimation error typically results in imprecise guidance gradient, leading to inconsistent generation outcomes.
- We advocate for the dual-focus evaluation framework to better assess guidance-based methods and reveal the severe issue of cross-condition interference in existing methods.
- We analyze the source of the estimation error and propose a simple, plug-and-play improvement strategy ABMS to mitigate its impact and also provide theoretical support.
- Experiments on various tasks and data types demonstrate the generality and effectiveness of the proposed strategy.

## 2 RELATED WORK

**Training-free conditional diffusion sampling.** In recent years, the technology of training-free conditional sampling in diffusion models (Wang et al., 2023; Kawar et al., 2022; Chung et al., 2023; 2022; Lugmayr et al., 2022) has witnessed rapid development (Dou & Song, 2024; Cardoso et al., 2024; Feng et al., 2023; Janati et al., 2024). Essentially, these methods enable conditional generation under any differentiable constraint. For example, Red-Diff (Mardani et al., 2023) casts the condition as a differentiable loss and treats the diffusion prior as a regularizer, thereby formulating the problem as an optimization task. Another popular and widely adopted paradigm is Diffusion Posterior Sampling (DPS)(Chung et al., 2023), which is the basic of our method.

As mentioned earlier, the core idea of DPS is to leverage Tweedie's formula to approximate conditional scores, generalizing linear inverse problem solvers to arbitrary generation tasks. In a series of subsequent works on DPS, LGD-MC (Song et al., 2023) reduces estimation bias by using Monte Carlo sampling from imperfect Gaussian distributions. However, the strong assumption that $p(x_0|x_t)$ follows a Gaussian distribution prevents any increase in sampling steps from adequately capturing the multi-modal distributions in real world scenarios. Other work such as UGD (Bansal et al., 2023) and DiffPIR (Zhu et al., 2023) perform guidance on clean data $x_0$ followed by projection to intermediate states $x_t$. Recently, MPGD (He et al., 2024) enforces guidance within the tangent space of the clean data manifold using autoencoders, improving sample quality at the cost of relying on pre-trained models and linearity assumptions. DSG (Yang et al., 2024), on the other hand, focuses on the high-density regions near the center of the high-dimensional Gaussian distribution, constraining the guidance step within the intermediate data manifold through optimization, and enabling the use of larger guidance steps. However, it is worth noting that the existing improvements to DPS primarily focus on preventing the intermediate data $x_t$ from deviating from the manifold $\mathcal{M}_t$ at time step $t$, without addressing the issue of the *guidance gradient imprecision* in the plain DPS.

**Inference time scale diffusion sampling.** Several existing methods enhance sample quality without modifying model parameters by leveraging additional computational cost during sampling(Ma et al., 2025; Zhang et al., 2025; Xu et al., 2023). These methods typically fully exploit the capability of pretrained generative models through noise search or reintroducing noise during the generation process to restart sampling. Motivated by the same principle, our method aims to obtain more accurate guidance gradients for arbitrary differentiable conditions by increasing appropriate computational budget.

## 3 PRELIMINARY

### 3.1 DIFFUSION MODELS FUNDAMENTALS

Diffusion models (Sohl-Dickstein et al., 2015) construct a forward process that gradually adds noise to data samples through $T$ time steps, then learn to reverse this process for generation. The framework has evolved through two main perspectives: stochastic differential equations (SDEs) (Song et al., 2021b) and discrete-time Markov chains (Ho et al., 2020).

**Stochastic differential equation framework.** The continuous-time perspective formulates diffusion through SDEs:

$$d\boldsymbol{x} = h(\boldsymbol{x}, t)dt + g(t)\boldsymbol{w}. \tag{1}$$

where $h(\cdot)$ is the drift coefficient, $g(t)$ controls noise scaling, and $\boldsymbol{w}$ denotes Brownian motion (Anderson, 1982). The reverse process becomes:

$$d\boldsymbol{x} = [h(\boldsymbol{x}, t) - g(t)^2 \nabla_{\boldsymbol{x}} \log p_t(\boldsymbol{x})]dt + g(t)d\bar{\boldsymbol{w}}. \tag{2}$$

where $\nabla_{\boldsymbol{x}} \log p_t(\boldsymbol{x})$ is estimated by a neural network. Various noise schedules yield distinct forward-noising and reverse trajectories but all are subsumed under Equation 2.

### 3.2 TRAINING-FREE GUIDED DIFFUSION

**Problem formulation.** Let $\boldsymbol{x}_0 \in \mathcal{X} \subseteq \mathbb{R}^d$ be the *unknown* clean signal we wish to generate or recover, and $\boldsymbol{y} \in \mathcal{Y}$ an *arbitrary conditioning variable* (e.g. a corrupted measurement, categories, or numerical properties). We assume access to

- a pre-trained diffusion prior$p(\boldsymbol{x}_0)$ that models the marginal distribution of clean data;
- a differentiable, plug-and-play loss $\mathcal{L}(\boldsymbol{x}_0; \boldsymbol{y})$ defined *on the clean space $\mathcal{X}$*, which encodes the conditional likelihood $p(\boldsymbol{y}|\boldsymbol{x}_0) \propto \exp\big(-\mathcal{L}(\boldsymbol{x}_0; \boldsymbol{y})\big)$.

Our goal is to sample from the posterior

$$p(\boldsymbol{x}_0|\boldsymbol{y}) \propto p(\boldsymbol{x}_0) \exp\big(-\mathcal{L}(\boldsymbol{x}_0; \boldsymbol{y})\big).$$

No retraining or time-dependent classifier is required: guidance is performed by back-propagating the loss $\mathcal{L}$ at every diffusion time-step, yielding a fully training-free inference pipeline.

**Diffusion posterior sampling.** Diffusion posterior sampling (DPS) (Chung et al., 2023) casts deterministic inverse problems as posterior sampling via the Bayesian identity:

$$\nabla_{\boldsymbol{x}_t} \log p(\boldsymbol{x}_t|\boldsymbol{y}) = \nabla_{\boldsymbol{x}_t} \log p(\boldsymbol{x}_t) + \underbrace{\nabla_{\boldsymbol{x}_t} \log p(\boldsymbol{y}|\boldsymbol{x}_t)}_{\text{intractable}}. \tag{3}$$

To sidestep the intractable likelihood score, DPS performs *two successive approximations*. **Approximation-1:** Factories the likelihood and move the expectation *inside* the loss,

$$p(\boldsymbol{y}|\boldsymbol{x}_t) = \int p(\boldsymbol{y}|\boldsymbol{x}_0) \, p(\boldsymbol{x}_0|\boldsymbol{x}_t) \, d\boldsymbol{x}_0 \ \approx \ p(\boldsymbol{y}|\mathbb{E}[\boldsymbol{x}_0|\boldsymbol{x}_t]). \tag{4}$$

**Approximation-2:** Invoke Tweedie's formula and replace the posterior mean with the *plug-in* estimate produced by the denoising network $\hat{\boldsymbol{x}}_0$. Combining the two steps yields the gradient surrogate:

$$\nabla_{\boldsymbol{x}_t} \log p(\boldsymbol{y}|\boldsymbol{x}_t) \ \approx \ \nabla_{\boldsymbol{x}_t} \log p(\boldsymbol{y}|\hat{\boldsymbol{x}}_0(\boldsymbol{x}_t)). \tag{5}$$

## 4 METHOD

In this part, we first analyze the limitations of current mainstream approaches in Section 4.1 and propose mitigation strategies. Then, in Section 4.2, we conduct an error analysis, providing a theoretical guarantee for the effectiveness of the proposed strategy. Finally, in Section 4.3, we present the complete pipeline of the method.

### 4.1 ADDITIONAL BACKWARD STEP WITH MONTE-CARLO SAMPLER

**The estimation error problem in DPS.** As discussed in Section 3.2, more generally, for any differentiable conditional function $f(x)$, with the conditioning variable $y$ suppressed for notational simplicity: **the core objective** is to estimate $\mathbb{E}_{x_0|x_t}[f(x_0)]$, *i.e.*, the conditional expectation of $f$ evaluated at the clean signal $x_0$, given the noisy input $x_t$. The common approach DPS uses the denoising network output $\hat{x}_0(x_t)$ to predict $x_0$ and uses the conditional gradient to guide the diffusion generation process as:

$$x_{t-1} \leftarrow x'_{t-1} - \omega_t \nabla_{x_t} f(\hat{x}_0(x_t)), \tag{6}$$

where $x'_{t-1}$ is the unconditional update term and $\omega_t$ is the guidance scale. However, this single-point approximation fails to account for the inherent uncertainty in $p(x_0|x_t)$ and introduces significant bias, particularly when $f$ is nonlinear (by Jensen's inequality) and $x_t$ is highly noisy. Crucially, while the target quantity $\mathbb{E}_{x_0|x_t}[f(x_0)]$ depends on the full posterior $p(x_0|x_t)$, this distribution is generally complex and analytically intractable, precluding direct sampling or exact computation.

**Leveraging the trackable structure in diffusion backward process.** Fortunately, we notice that the reverse diffusion process defines a Markov chain in which the one-step transition kernel $p(x_{t-1}|x_t)$ can be substituted by an explicitly parameterized Gaussian in practice. This allows us to re-express the desired expectation via the law of total expectation:

$$\mathbb{E}_{x_0|x_t}[f(x_0)] = \mathbb{E}_{x_{t-1}|x_t}\left[\mathbb{E}[f(x_0)|x_{t-1}]\right].$$

Rather than approximating $x_0$ directly from $x_t$, our method propagates uncertainty through an intermediate step, effectively averaging over multiple plausible denoising paths. Based on this insight, we define the **ABMS** strategy as follows:

1. Sample $M$ intermediate states: $x_{t-1}^{(m)} \sim p(x_{t-1}|x_t)$ for $m = 1, \ldots, M$.

2. For each $x_{t-1}^{(m)}$, obtain a denoised estimate $\hat{x}_0(x_{t-1}^{(m)})$ using the pretrained denoising network.

3. Evaluate the conditional function: compute $f(\hat{x}_0(x_{t-1}^{(m)}))$ for each sample.

4. Average the evaluations:

$$\hat{f}_{\text{ABMS}}(M, x_t) = \frac{1}{M} \sum_{m=1}^{M} f(\hat{x}_0(x_{t-1}^{(m)})). \tag{7}$$

Intuitively, by injecting a stochastic intermediate step, ABMS lets the generative network explore multiple plausible denoising trajectories, naturally capturing the multi-modal shape of $p(x_0|x_t)$ instead of being trapped in a single point estimation. Furthermore, in the experiment section we also demonstrate the method's suitability for higher order samplers involving stochasticity, not just for single step updates.

### 4.2 ESTIMATION ERROR ANALYSIS

We also present a rigorous comparison of the estimation error between the DPS estimator and our ABMS estimator. To establish our theoretical results, we make the following assumptions, which are empirically well-supported and commonly adopted in diffusion modeling:

**A1.** The conditional function $f : \mathcal{X} \to \mathbb{R}^d$ is $K$-Lipschitz continuous and $L$-Lipschitz gradient.

**A2.** The accuracy of the denoiser improves monotonically along the reverse diffusion process. Specifically, for any state $x_t$, the expected reconstruction error satisfies:

$$\mathbb{E}_{x_{t-1}|x_t}\left[\left\|\hat{x}_0(x_{t-1}) - \mathbb{E}_{x_0|x_{t-1}}[x_0]\right\|\right] \leq \left\|\hat{x}_0(x_t) - \mathbb{E}_{x_0|x_t}[x_0]\right\|.$$

This assumption reflects the intuitive and empirically observed fact that reconstructions from less noisy intermediate states ($x_{t-1}$) are closer to the true posterior mean than those from noisier states ($x_t$).

Then we can have the following proposition:

> *Proposition 1: ABMS attains a lower-bounded expected estimation error than plain DPS.*

*Proof.* **Error bound for DPS.** The DPS estimator uses a single reconstruction $\hat{x}_0(x_t)$ to approximate $\mathbb{E}_{x_0|x_t}[f(x_0)]$, yielding:

$$\hat{f}_{\text{DPS}} = f(\hat{x}_0(x_t)).$$

Its estimation error is bounded by decomposing:

$$\begin{aligned}
\|\text{Error}_{\text{DPS}}\| &= \left\|f(\hat{x}_0(x_t)) - \mathbb{E}_{x_0|x_t}[f(x_0)]\right\| \\
&\leq \left\|f(\hat{x}_0(x_t)) - f(\mathbb{E}_{x_0|x_t}[x_0])\right\| + \underbrace{\left\|f(\mathbb{E}_{x_0|x_t}[x_0]) - \mathbb{E}_{x_0|x_t}[f(x_0)]\right\|}_{\text{Jensen gap item:}\delta_f(x_t)} \\
&\leq K \cdot \left\|\hat{x}_0(x_t) - \mathbb{E}_{x_0|x_t}[x_0]\right\| + \delta_f(x_t),
\end{aligned} \tag{8}$$

where $\delta_f(x_t) \geq 0$ known as Jensen gap quantifies the deviation due to the nonlinearity of $f$, and vanishes only if $f$ is affine.

**Error bound for ABMS.** Recalling the ABMS estimator defined in equation 7, as $M$ is large enough, it converges to $\mathbb{E}_{x_{t-1}|x_t}[f(\hat{x}_0(x_{t-1}))]$. Using the Markov property,

$$\mathbb{E}_{x_0|x_t}[f(x_0)] = \mathbb{E}_{x_{t-1}|x_t}\left[\mathbb{E}_{x_0|x_{t-1}}[f(x_0)]\right],$$

the asymptotic estimation error satisfies:

$$\|\text{Error}_{\text{ABMS}}\| = \mathbb{E}_{x_{t-1}|x_t}\left[\left\|f(\hat{x}_0(x_{t-1})) - \mathbb{E}_{x_0|x_{t-1}}[f(x_0)]\right\|\right] \tag{9}$$

$$\leq K \cdot \mathbb{E}_{x_{t-1}|x_t}\left[\left\|\hat{x}_0(x_{t-1}) - \mathbb{E}_{x_0|x_{t-1}}[x_0]\right\|\right] + \mathbb{E}_{x_{t-1}|x_t}[\delta_f(x_{t-1})].$$

**Comparison of expected error bounds.** Taking expectation over $x_t \sim p(x_t)$ on both equation 8 and equation 9 yields:

$$\mathbb{E}\|\text{Error}_{\text{DPS}}\| \leq K\,\mathbb{E}\|\hat{x}_0(x_t) - \mathbb{E}_{x_0|x_t}[x_0]\| + \mathbb{E}[\delta_f(x_t)], \tag{10}$$

$$\mathbb{E}\|\text{Error}_{\text{ABMS}}\| \leq K\,\mathbb{E}\left[\mathbb{E}_{x_{t-1}|x_t}\|\hat{x}_0(x_{t-1}) - \mathbb{E}_{x_0|x_{t-1}}[x_0]\|\right] + \mathbb{E}\left[\mathbb{E}_{x_{t-1}|x_t}[\delta_f(x_{t-1})]\right]. \tag{11}$$

**Step 1: reconstruction term.** By Assumption 2, we have:

$$\mathbb{E}_{x_{t-1}|x_t}\|\hat{x}_0(x_{t-1}) - \mathbb{E}_{x_0|x_{t-1}}[x_0]\| \leq \|\hat{x}_0(x_t) - \mathbb{E}_{x_0|x_t}[x_0]\|,$$

so the first term in equation 10 which related to reconstruction error is larger than that in equation 11.

**Step 2: Jensen-gap term.** According to the $L$-Lipschitz gradient property in Assumption 1, we can get the upper bound of $\mathbb{E}[\delta_f(x_t)]$ and $\mathbb{E}\left[\mathbb{E}_{x_{t-1}|x_t}[\delta_f(x_{t-1})]\right]$ respectively (the proof is provided in Appendix A.1):

$$\begin{aligned}
\text{UB}_t &= \tfrac{1}{2}L\,\mathbb{E}_{x_t}\left[\text{Tr}\left(\text{Cov}_{x_0|x_t}[x_0]\right)\right], \\
\text{UB}_{t-1} &= \tfrac{1}{2}L\,\mathbb{E}_{x_t}\left[\mathbb{E}_{x_{t-1}|x_t}\text{Tr}\left(\text{Cov}_{x_0|x_{t-1}}[x_0]\right)\right].
\end{aligned} \tag{12}$$

Law of total covariance gives the identity:

$$\text{Cov}_{x_0|x_t}[x_0] = \mathbb{E}_{x_{t-1}|x_t}\left[\text{Cov}_{x_0|x_{t-1}}[x_0]\right] + \text{Cov}_{x_{t-1}|x_t}\left[\mathbb{E}_{x_0|x_{t-1}}[x_0]\right].$$

Taking trace and expectation over $x_t$ and we find that:

$$\text{UB}_t - \text{UB}_{t-1} = \tfrac{1}{2}L\,\mathbb{E}_{x_t}\left[\text{Tr}\left(\text{Cov}_{x_{t-1}|x_t}\left[\mathbb{E}_{x_0|x_{t-1}}[x_0]\right]\right)\right] \geq 0.$$

Hence $\text{UB}_{t-1} \leq \text{UB}_t$, showing that the Jensen gap in ABMS contributes to the expected error upper bound is no larger than that of DPS. Combining the two steps, we have proved the original proposition. $\square$

### 4.3 THE COMPLETE FRAMEWORK OF ABMS GUIDANCE

In the preceding sections we introduced the ABMS estimator to obtain a more accurate guidance direction, denoted as $g$. As regards the scale of the guidance, recall that the posterior $p(x_{t-1}|x_t)$ follows a Gaussian distribution: $p(x_{t-1}|x_t) \sim \mathcal{N}(\mu_\theta(x_t, t), \sigma_t^2 I)$. Inspired by DSG (Yang et al., 2024), for any $n$-dimensional isotropic Gaussian $x \sim \mathcal{N}(\mu, \sigma^2 I)$, when $n$ is sufficiently large the distribution is concentrated on a hypersphere of radius $\sqrt{n}\sigma$ centred at $\mu$. To prevent the guided sample from drifting away from the data manifold, we therefore rescale and constrain the magnitude of the guidance vector to lie in this hypersphere: $g' = \omega_t \cdot \sqrt{n}\sigma_t \cdot \frac{g}{\|g\|}$, where $\omega_t \in (0, 1)$ is the guidance rate. We employ a cosine schedule to ensure that $\omega_t$ increases smoothly throughout the denoising process and the complete procedure is provided in Algorithm 1.

---

**Algorithm 1** One Guided Diffusion Step of ABMS

---

**Require:** $x_t, x_{t-1}^{\text{mean}}, \sigma_t, w_{\max}, f, M, T, n$
1: radius $\leftarrow \sqrt{n} \cdot \sigma_t$          $\triangleright$ $n$ denotes dimensionality
2: $\mathcal{F} \leftarrow \emptyset$
3: **for** $i = 1$ **to** $M$ **do**          $\triangleright$ draw $M$ Monte-Carlo samples
4:     $\varepsilon_t \sim \mathcal{N}(\mathbf{0}, I)$
5:     $x_{t-1} \leftarrow x_{t-1}^{\text{mean}} + \sigma_t \cdot \varepsilon_t$
6:     $\hat{x}_0 \leftarrow \text{pred\_x0}(x_{t-1})$          $\triangleright$ network predicts clean $x_0$
7:     $\mathcal{F} \leftarrow \mathcal{F} \cup \{f(\hat{x}_0)\}$
8: **end for**
9: $\hat{f} \leftarrow \text{average}(\mathcal{F}), g \leftarrow -\nabla_{x_t}\hat{f}$          $\triangleright$ ABMS guidance direction
10: $\omega_t = \frac{w_{\max}}{2}\left(1 + \cos\left(\pi(1 - t/T)\right)\right)$
11: $g' \leftarrow \omega_t \cdot \text{radius} \cdot \frac{g}{\|g\|}$
12: $x_{t-1}^{\text{new}} \leftarrow x_{t-1}^{\text{mean}} + g' + \sigma_t \cdot \varepsilon_t$
**Ensure:** $x_{t-1}^{\text{new}}$

---

## 5 EXPERIMENTS

In this section, we evaluate our methods extensively in various tasks, including a dual-conditional generation task (stylized handwritten trajectory generation), three prevalent image inverse problems (inpainting, super resolution and gaussian deblurring), molecular inverse design and text-style guidance. We primarily compare with the current state-of-the-art method, DSG (Yang et al., 2024), which has been proven to effectively prevent the manifold deviation phenomena.

A key aspect of our experimental design is the dual-focus evaluation criterion: simultaneously assessing (1) the consistency of the generated samples with the specified conditions, and (2) the resulting impact on other sample attributes. Below we detail the metric pairs used in each experiment: **content score** vs **style score** in Section 5.1, **distance** vs **FID** in Section 5.2, and **MAE** vs **mol stability** in Section 5.3. For the analysis of sampling time, please refer to Appendix A.2. For more experiment results, please refer to Appendix A.3.

### 5.1 STYLIZED HANDWRITTEN CHARACTER GENERATION

**Implementation.** The goal of this task is to generate Chinese characters with specified *categories* and *writing styles*. Following the setup in (Ren et al., 2025), we use the pretrained diffusion model with dual-conditional generation capabilities. We use the CASIA-OLHWDB (1.0-1.2) dataset (Liu et al., 2011) as the training set and the ICDAR-2013 competition database (Yin et al., 2013) as the test set. We adopt the DDPM sampler with 1000 sampling steps.

For evaluation, we train two classifiers for character categories and handwriting styles on the test set respectively, following the previous work (Ren et al., 2025; Dai et al., 2023). The classification accuracy of the synthesized samples is called the ***content score*** and the ***style score***. Higher scores indicate that the synthesized samples are closer to the real target data.

**Cross-condition interference.** In an ideal scenario (*i.e.,* on a clean data manifold), the category and writing style are completely decoupled conditions. Therefore, applying a moderate gradient to

Table 1: Quantitative evaluation of conditional character generation under different guidance scales.

| Method-Scale | No guidance | DSG(0.01) | DSG(0.1) | DSG(0.5) | Ours(0.01) | Ours(0.1) | Ours(0.5) |
|---|---|---|---|---|---|---|---|
| Content($\uparrow$) | 0.827 | 0.927 | 0.998 | 0.999 | 0.981 | 0.999 | 0.999 |
| Style($\uparrow$) | 0.899 | 0.543 | 0.534 | 0.166 | 0.888 | 0.878 | 0.756 |

one condition should not have a negative impact on the other. To this end, we use *only the gradient from category classifier* to guide the diffusion model, while observing any potential impact on the style score. We evaluate performance under varying guidance scales and the number of sampling time $M$ is set to 3 in our ABMS method.

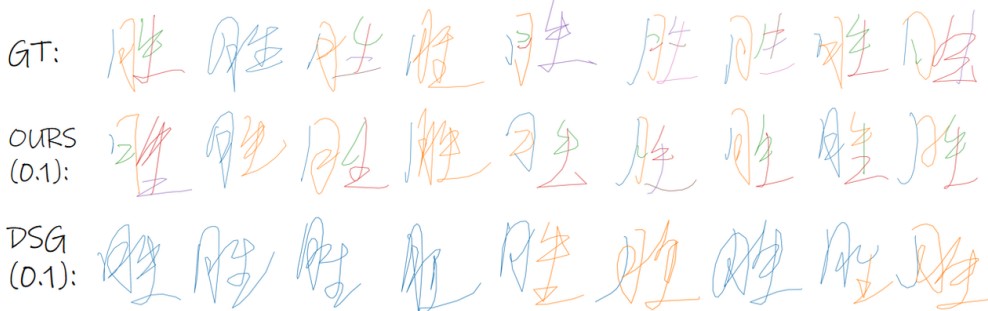

Figure 1: The visualization comparison results, where different colors represent different strokes. The guidance scale is set as 0.1. It can be observed that, even without manifold deviation, the fonts generated by DSG tend to have **connected strokes**, regardless of the target writing style. On the other hand, our method is able to better preserve the style characteristics.

**Evaluation results.** As shown in Table 1, both the two methods can significantly boost the content score. However, even with a smaller guidance scale, DSG method based on plain DPS gradients still exerts a significant influence on style features. In contrast, our method shows substantial improvements in both quantitative and qualitative evaluations. Figure 1 demonstrates that at a guidance scale of 0.1, our proposed method basically maintains the writing style of the synthesized stroke trajectories, while DSG introduces pronounced stylistic distortions in calligraphic consistency.

## 5.2 Image Inverse Problems

**Implementation.** In this part, we investigate three prevalent linear inverse problems of the form $y = Ax + \epsilon$, which including **image inpainting**, **super-resolution**, and **gaussian deblurring**. We evaluate our proposed method on two datasets: 1K images from FFHQ 256×256 (Karras et al., 2019) and ImageNet 256×256 validation dataset (Deng et al., 2009), using pre-trained diffusion models sourced from (Chung et al., 2023; Dhariwal & Nichol, 2021). For comparison, we select several baseline methods: DPS (Chung et al., 2023), LGD (Song et al., 2023) and DSG (Yang et al., 2024). We adopt the experimental setup from DSG to generate noisy measurements. The loss function guiding the reconstruction process is defined as follows:

$$\mathcal{L}(x_0, y) = \|A\hat{x}_0(x_t) - y\|_2^2, \tag{13}$$

where $y$ represents the noisy measurement, $A$ represents the deterioration model and $x_0$ denotes the image we aim to reconstruct. For these tasks, we employ DDIM Song et al., 2021a sampler with 100 sampling steps.

For evaluation, as shown in Equation 13, this loss function essentially represents the pixel-wise difference between the generated image and the real ground-truth image after both undergo the same degradation process. We refer to this as the '**Distance**' metric, which is used to evaluate the degree to which the guidance method adheres to the conditions. Additionally, we evaluated image quality metrics including PSNR, SSIM, LPIPS, and FID.

**Evaluation results.** The quantitative evaluation results for the tasks mentioned above are presented in Table 2. Notably, our method demonstrates consistent performance improvements over baselines across a range of evaluation metrics and task configurations. These results provide empirical support for the feasibility of our proposed modification.

More importantly, following the principle of ***dual-focus evaluation***, we measure the performance curves of the Distance vs FID metrics under different guidance scales, as illustrated in Figure 2. Our method achieves a lower Distance while maintaining higher image quality. Due to space limitations, the results for the Gaussian Deblurring task are provided in the Appendix A.3 for reference.

For the case $M = 1$, the resulting curve is visually relatively similar to the initial DSG and is therefore omitted for clarity. It is noteworthy that as the number of sampling steps (see Equation 7) increases, the method exhibits relatively better performance, which aligns with the algorithm's intuition. Moreover, we observe that the performance gain becomes evident once $M$ reaches 3, and the marginal benefit gradually saturates as $M$ is further increased.

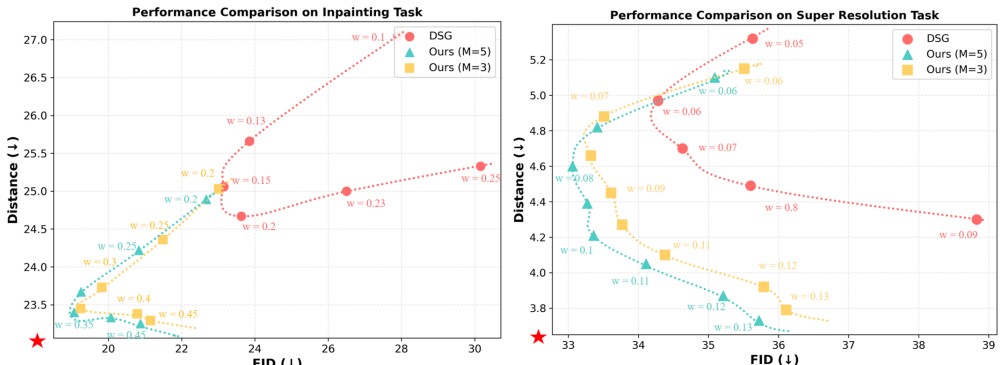

Figure 2: Performance curves of Distance Metric vs FID Metric. We select different guidance scales for each method to obtain performance trend curves. It can be clearly observed that our method achieves a better guidance effect and exhibits greater robustness to the selection of guidance scale.

Table 2: Quantitative image-quality evaluation on Imagnet under the linear inverse problem.

| Methods | Inpainting | | | | Super resolution | | | | Gaussian deblurring | | | |
|---|---|---|---|---|---|---|---|---|---|---|---|---|
| | PSNR↑ | SSIM↑ | LPIPS↓ | FID↓ | PSNR↑ | SSIM↑ | LPIPS↓ | FID↓ | PSNR↑ | SSIM↑ | LPIPS↓ | FID↓ |
| DPS | 27.56 | 0.825 | 0.251 | 30.57 | 22.07 | 0.633 | 0.317 | 41.36 | 18.78 | 0.387 | 0.523 | 52.13 |
| LGD | 27.78 | 0.824 | 0.242 | 28.65 | 22.23 | 0.631 | 0.305 | 39.85 | 19.52 | 0.452 | 0.501 | 50.42 |
| DSG | 28.67 | 0.877 | 0.176 | 23.63 | 23.74 | 0.696 | 0.291 | 34.28 | 22.64 | 0.644 | 0.335 | 45.27 |
| Ours | **29.23** | **0.889** | **0.154** | **19.25** | **23.80** | **0.697** | **0.290** | **33.06** | **22.65** | **0.645** | **0.321** | **41.65** |

### 5.3 MOLECULAR INVERSE DESIGN

**Implementation.** Inverse molecular design aims to generate 3D molecular structures that exhibit desired quantum properties. Following the settings in EDM (Hoogeboom et al., 2022) and EEGSDE (Bao et al., 2022), we adopt the QM9 (Ramakrishnan et al., 2014) dataset, which contains ∼130k organic molecules with up to nine heavy atoms (C, N, O, F) and their corresponding quantum properties. Six properties are selected: polarizability ($\alpha$), dipole moment ($\mu$), HOMO/LUMO energies ($\varepsilon_{\text{HOMO}}$, $\varepsilon_{\text{LUMO}}$), HOMO-LUMO gap ($\Delta\varepsilon$), and heat capacity at constant volume ($C_v$). All experiments are conducted with the pre-trained EEGSDE model. More specifically, for each property, two separate predictors are needed: (i) a time-dependent, noise-robust predictor $g_\phi(x_t, t)$ used by EEGSDE to guide the diffusion process, and (ii) an evaluation predictor $\varphi_p(x_0)$, which has been trained on clean data only used to assess the generated molecules. In Table 3, "L-bound" represents the MAE error of the evaluation predictor on the test data.

The baseline conditional EDM (cEDM) receives the desired property as an additional input during training to learn a conditional generative model without guidance. Based on EDM, EEGSDE incor-

porates an equivariant energy function $E(x_t, c, t) = s(g_\phi(x_t, t) - c)^2$ into the reverse-time SDE, where $s$ is a tunable scale and $c$ is the desired quantum property. For EEGSDE, since $g_\phi$ must be robust to noise, it is conditioned on both the perturbed coordinates $x_t$ and the time step $t$. In contrast, our training-free approach only requires a predictor defined on clean data ($t = 0$). However, *for a fair comparison, we reuse the EEGSDE-trained $g_\phi$*, but freeze $t = 0$ when providing guidance.

Table 3: Quantitative Evaluation of Molecular Property Prediction. MAE represents the numerical discrepancy between the specific properties of generated molecules and the target conditions, while MS denotes the molecular stability index.

| Method | MAE ($\downarrow$) | MS ($\uparrow$) | Method | MAE ($\downarrow$) | MS ($\uparrow$) | Method | MAE ($\downarrow$) | MS ($\uparrow$) |
|---|---|---|---|---|---|---|---|---|
| | $C_v$ | | | $\mu$ | | | $\alpha$ | |
| CEDM | 1.0650 | – | CEDM | 1.1230 | – | CEDM | 2.7804 | – |
| EEGSDE | 0.9187 | 0.7836 | EEGSDE | 0.7518 | 0.8036 | EEGSDE | 2.4133 | 0.7954 |
| DSG | 0.8447 | 0.7654 | DSG | 0.7811 | 0.8001 | DSG | 2.1919 | 0.7863 |
| Ours($M = 3$) | **0.8348** | 0.7718 | Ours($M = 3$) | **0.7274** | 0.8059 | Ours($M = 3$) | **2.1001** | 0.7745 |
| L-bound | 0.0400 | – | L-bound | 0.0430 | – | L-bound | 0.0900 | – |
| | $\triangle\epsilon$ | | | $\epsilon_{HOMO}$ | | | $\epsilon_{LUMO}$ | |
| CEDM | 0.6710 | – | CEDM | 0.3714 | – | CEDM | 0.6015 | – |
| EEGSDE | 0.4854 | 0.7572 | EEGSDE | 0.2997 | 0.7890 | EEGSDE | 0.4450 | 0.8000 |
| DSG | 0.4558 | 0.7890 | DSG | 0.2673 | 0.7854 | DSG | 0.3969 | 0.8081 |
| Ours($M = 3$) | **0.4182** | 0.7909 | Ours($M = 3$) | **0.2449** | 0.7872 | Ours($M = 3$) | **0.3778** | 0.8181 |
| L-bound | 0.0650 | – | L-bound | 0.0390 | – | L-bound | 0.0360 | – |

For evaluation, we still adhere to the principle of ***dual-focus evaluation***, assessing both the metric of Molecular Property Deviation from Conditions (MAE) and the Molecular Stability (MS) metric. More specifically, we observe that for each method, as the guidance scale increases, the MAE metric gradually decreases, but this may be accompanied by a decline in the MS metric. Therefore, we first reproduced the parameters of EEGSDE, and then adjusted the guidance scales of the DSG and ABMS methods until their MS metrics either outperformed EEGSDE's or differed by no more than 2% before comparing their MAE metrics.

**Evaluation Results.** As shown in Table 3, under the condition of comparable molecular stability (MS), our method achieves superior MAE metrics across six distinct conditional molecular inverse design tasks. This further demonstrates that our method can provide more accurate conditional guidance compared to existing methods. It can be observed that tasks conditioned on precise numerical values exhibit significantly degraded performance when guidance is absent, corroborating the necessity of explicit guidance mechanisms. Moreover, we emphasize that our framework only requires a property predictor defined on clean data; for fair comparison, we reused the noise-robust predictor trained by EEGSDE, which may partially limit the achievable performance.

## 5.4 SCALING UP TEXT STYLE GUIDANCE

To thoroughly validate the generalizability of our method, We conduct experiments on diffusion model of **larger scale** and **distinct training strategy**. Specifically, we perform text-style guidance task and adopt Stable Diffusion 3.5 (Rombach et al., 2022; Esser et al., 2024) as the generative diffusion prior, which is a \*\**flow matching*\*\* Lipman et al., 2023; Liu et al., 2023 based model and receives a piece of text description as input. We adopt the method proposed in (Liu et al., 2025) to enable SDE sampling. We apply the style loss for guidance, which is defined as:

$$\mathcal{L}(\hat{x}_0(x_t), x_{in}) = \|E(\hat{x}_0(x_t)) - E(x_{in})\|_2^F, \tag{14}$$

where $x_{in}$ is the style reference image, $E$ represents the Gram matrix of the third feature map extracted from the CLIP image encoder (Radford et al., 2021), and $\|\cdot\|_2^F$ denotes the Frobenius norm. The size of the generated images is 512×512. Figures 3 presents the visual results of our method with $M = 3$ compared to the baseline method. As can be seen, our method also achieves satisfying performance on flow matching based models and produces clearer and much higher image quality while ensuring adherence to the conditional guidance. This is because more accurate guidance

gradients enable the generation process to satisfy the conditions as much as possible while not undermining the prior diffusion capabilities.

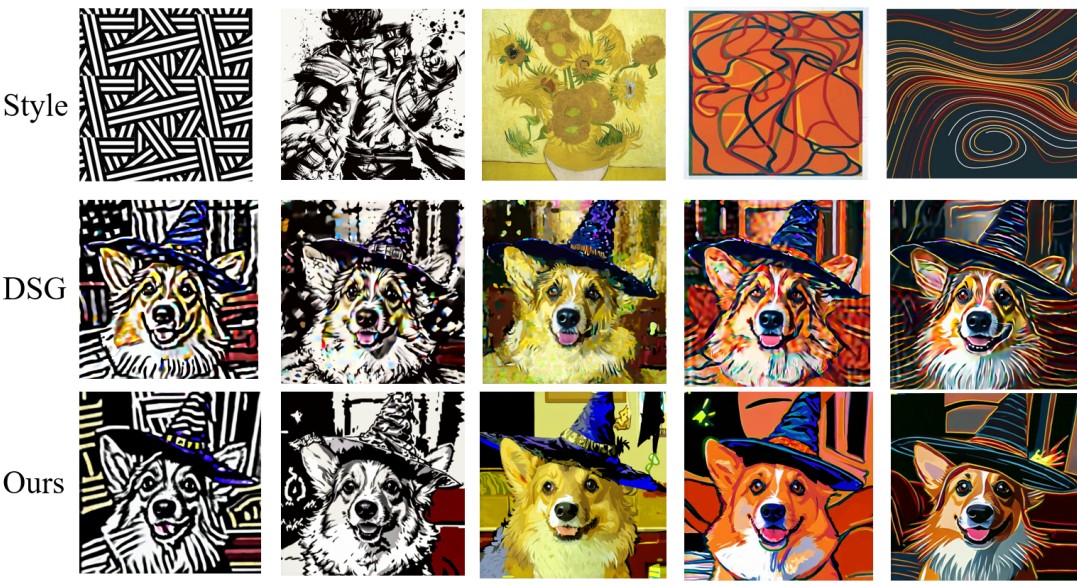

Figure 3: Qualitative result of Text-style guidance, the text input is "A corgi wearing a wizard hat". Our method generates much clearer and higher quality results than the baseline method across all the target style.

## 6  CONCLUSION & DISCUSSION

In this paper, we propose a simple plug-and-play strategy ABMS to improve the existing DPS-based guidance methods and a dual-focus evaluation paradigm to offer a more discerning assessment of guidance methodologies. To mitigate the giant estimation error when calculating the guidance gradient, we demonstrate that the strategy of performing an additional denoising step with Monte-Carlo sampling can reduce the variance interference, effectively improves the performance.

**Limitation and future work.** Although the proposed method has demonstrated its efficacy in integrating and enhancing existing approaches, constrained by computational resources, we have yet to systematically investigate whether further increasing the number of reverse diffusion steps or the sampling budget yields additional gains across diverse scenarios. Additionally, how the proposed methodology can be adapted to paradigms that enable very few-step generation is also an open question. These will be exciting directions for future work.

## 7  ETHICS STATEMENT.

This paper focuses on the research of general artificial intelligence technology, specifically the generative diffusion models, and does not involve ethic issues.

## 8  REPRODUCIBILITY STATEMENT.

For a fair comparison, all the experiments mentioned strictly follow the framework, data, and pre-trained diffusion models from existing articles, with corresponding references provided in the main text.

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

# A APPENDIX

## A.1 ADDITIONAL PROOF

**Lemma 1** (Jensen Gap Upper Bound). *Let $f : \mathbb{R}^d \to \mathbb{R}$ be a continuously differentiable function with L-Lipschitz gradient, i.e.,*

$$\|\nabla f(u) - \nabla f(v)\| \leq L\|u - v\| \quad \forall u, v \in \mathbb{R}^d. \tag{15}$$

*Then for any random variable $X$ with finite second moment, the Jensen gap satisfies:*

$$\delta_f(x) := f(\mathbb{E}[X|x]) - \mathbb{E}[f(X|x)] \leq \frac{L}{2} \operatorname{Tr}(\operatorname{Cov}[X|x]). \tag{16}$$

*Proof.* We proceed by Taylor expansion and the Lipschitz gradient property.

**Step 1: Second-order Taylor expansion.** Expanding $f$ around $\mathbb{E}[X|x]$ with remainder term:

$$f(X) = f(\mathbb{E}[X|x]) + \nabla f(\mathbb{E}[X|x])^\top (X - \mathbb{E}[X|x])$$
$$+ \int_0^1 (1 - s)(X - \mathbb{E}[X|x])^\top \nabla^2 f(\mathbb{E}[X|x] + s(X - \mathbb{E}[X|x]))(X - \mathbb{E}[X|x]) \, ds. \tag{17}$$

**Step 2: Taking conditional expectation.** Taking $\mathbb{E}[\cdot|x]$ on both sides of equation 17:

$$\mathbb{E}[f(X)|x] = f(\mathbb{E}[X|x]) + \nabla f(\mathbb{E}[X|x])^\top \mathbb{E}[X - \mathbb{E}[X|x]|x]$$
$$+ \mathbb{E}\left[\int_0^1 (1 - s)(X - \mathbb{E}[X|x])^\top \nabla^2 f(\cdot)(X - \mathbb{E}[X|x]) \, ds \,\Big|\, x\right]. \tag{18}$$

The second term vanishes since $\mathbb{E}[X - \mathbb{E}[X|x]|x] = 0$. Thus:

$$\delta_f(x) = f(\mathbb{E}[X|x]) - \mathbb{E}[f(X)|x] = -\mathbb{E}\left[\int_0^1 (1 - s)(X - \mathbb{E}[X|x])^\top \nabla^2 f(\cdot)(X - \mathbb{E}[X|x]) \, ds \,\Big|\, x\right]. \tag{19}$$

**Step 3: Bounding the Hessian norm.** The $L$-Lipschitz gradient property implies that for any $u, v \in \mathbb{R}^d$:

$$\|\nabla f(u) - \nabla f(v)\| \leq L\|u - v\|. \tag{20}$$

This ensures that the Hessian (in the sense of weak derivatives) satisfies the spectral bound:

$$\|\nabla^2 f(u)\| \leq L, \tag{21}$$

where $\|\cdot\|$ denotes the operator norm (largest eigenvalue).

**Step 4: Upper bounding the quadratic form.** For any vector $v \in \mathbb{R}^d$:

$$v^\top \nabla^2 f(u) v \leq \|\nabla^2 f(u)\| \|v\|^2$$
$$\leq L\|v\|^2. \tag{22}$$

Therefore:

$$(X - \mathbb{E}[X|x])^\top \nabla^2 f(\cdot)(X - \mathbb{E}[X|x]) \leq L\|X - \mathbb{E}[X|x]\|^2. \tag{23}$$

**Step 5: Taking expectation.** Applying conditional expectation:

$$\mathbb{E}\left[(X - \mathbb{E}[X|x])^\top \nabla^2 f(\cdot)(X - \mathbb{E}[X|x]) \,\Big|\, x\right] \leq L\,\mathbb{E}\left[\|X - \mathbb{E}[X|x]\|^2 \,\Big|\, x\right]$$
$$= L\operatorname{Tr}(\operatorname{Cov}[X|x]). \tag{24}$$

**Step 6: Final bound.** Substituting back into equation 19:

$$|\delta_f(x)| \le \mathbb{E}\left[\int_0^1 (1-s)L\|X - \mathbb{E}[X|x]\|^2\, ds \,\Big|\, x\right]$$

$$= L\,\mathrm{Tr}(\mathrm{Cov}[X|x])\int_0^1 (1-s)\, ds$$

$$= \frac{L}{2}\,\mathrm{Tr}(\mathrm{Cov}[X|x]). \tag{25}$$

This completes the proof. $\square$

### A.2 SAMPLING TIME OVERHEAD ANALYSIS

In practical applications, as shown in the experiment section we have found that a small number of Monte-Carlo sampling times already yields satisfactory results. To save computational time, we notice that computing the gradients with respect to multiple samples $x_{t-1}$ can be parallelized. We generally can choose the largest possible number of samples to ensure parallelization. In this case, Table 4 reports the number of diffusion-model denoising iterations completed per second under different values of $M$ for the image inverse problem.

In addition, a variety of acceleration techniques have been developed to speed up sampling in diffusion models (Song et al., 2021a; Lu et al., 2022; 2025; Lipman et al., 2023). Among them, DDIM and flow matching approaches can introduce stochasticity at every sampling step, we have therefore verified experimentally that our method remains compatible with these strategies, rendering the resulting computational overhead entirely acceptable.

Table 4: The inference time cost for different $M$.

| DPS/DSG | ABMS($M=1$) | ABMS($M=2$) | ABMS($M=3$) | ABMS($M=4$) | ABMS($M=5$) |
|---|---|---|---|---|---|
| 4.8 iter/s | 2.5 iter/s | 2.1 iter/s | 1.7 iter/s | 1.5 iter/s | 1.3 iter/s |

### A.3 ADDITIONAL EXPERIMENT RESULTS

In this section, we present some additional experimental results. As shown in Figure 4, the content guidance accurately corrects the trajectory structure of the generated characters, verifying the effectiveness of guidance method.

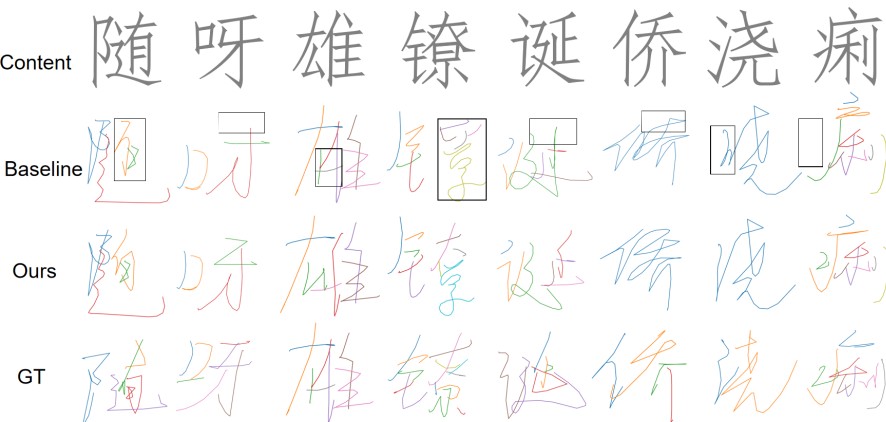

Figure 4: Qualitative result of content guidance. Compared to baseline model, the guidance we applied corrects the detailed strokes of the generated characters, making the character structures more accurate. The parts with structural errors are circled in a black box.

Figure 5 displays the experimental results for the Gaussian deblurring task discussed in the main text. We also provide the visualization of the linear inverse problems as shown in Figure 6. It can be observed that our improved strategy exhibits fewer artifacts and enhances the image quality.

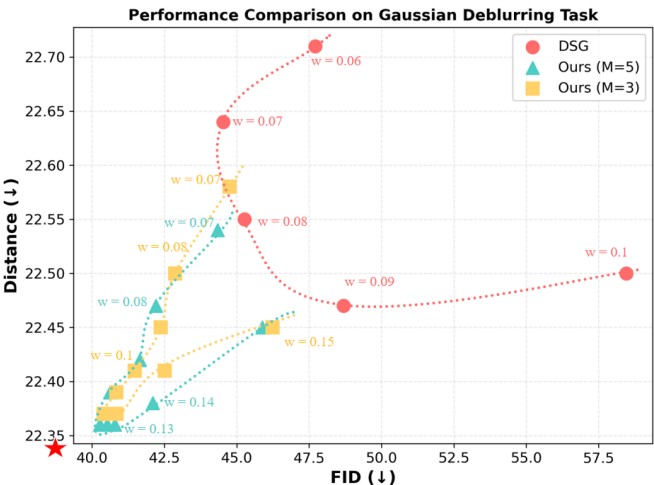

Figure 5: Performance curves of Distance Metric vs FID Metric on Gaussian Deblurring task.

We also evaluate our proposed methods on another nonlinear task: **FaceID guidance**. We utilize the pre-trained diffusion model from CelebA-HQ 256×256, provided by Freedom (Yu et al., 2023). We pick 1000 samples from the FFHQ test set (Karras et al., 2019) as the reference images. To guide the process for an input reference image $I$, we apply the FaceID loss, which is defined as:

$$\mathcal{L}(\hat{x}_0(x_t), I) = CE(E(\hat{x}_0(x_t)), E(I)), \tag{26}$$

where $E(\cdot)$ denotes the face recognition network (Deng et al., 2019) and $CE$ represents the Cross-Entropy Loss. We employ the DDIM sampler with 100 sampling steps.

Figures 7 presents the visual results of our method compared to other approaches. As can be seen, our method produces clearer details and higher image quality while ensuring adherence to the conditional guidance.

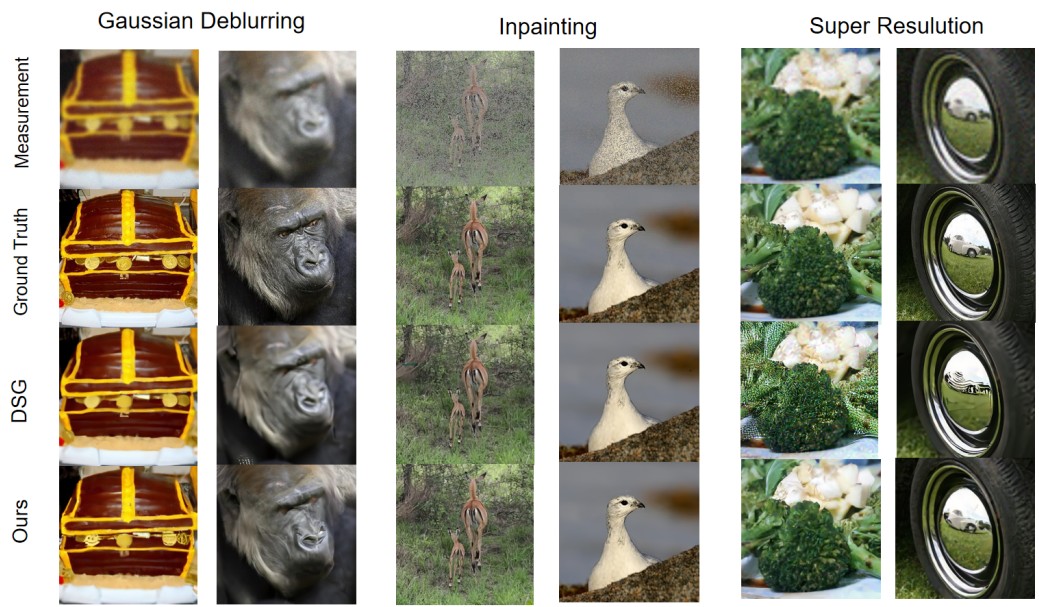

Figure 6: Qualitative result of linear inverse problems. Compared to existing methods, our improvements slightly enhance the image quality.

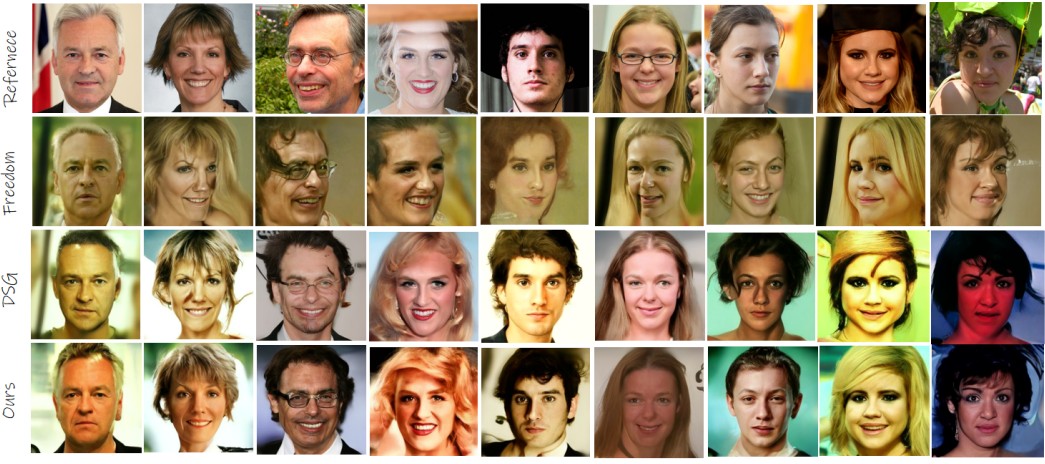

Figure 7: Qualitative result of FaceID guidance. Compared with existing methods, our improved approach generates more smooth results and is less likely to produce incoherent regions.

