# OpenReview forum: "One step further with Monte-Carlo sampler to guide diffusion better"
_ICLR.cc/2026/Conference — ICLR 2026 Poster_

### Official Review · Reviewer_Daim · 2025-10-18

**Soundness:** 2
**Presentation:** 2
**Contribution:** 2
**Rating:** 2
**Confidence:** 3

**Summary:**

The paper proposes ABMS for solving inverse problems using diffusion models.
It identifies and addresses errors arising from the imprecise gradient guidance in a prominent baseline method, DPS.
To resolve this, ABMS computes the gradient guidance by drawing multiple samples at each denoising step and using the diffusion model's predictions for those samples.
The proposed method shows enhanced performance on a variety of tasks, such as character generation, image restoration, and monocular property prediction.

**Strengths:**

- It highlights that many existing training-free inverse problem methodologies including DPS rely on numerous assumptions and approximations, which often leads to suboptimal results.

- It makes a significant contribution by demonstrating the trade-off inherent in using gradient guidance through a dual-focus evaluation.

**Weaknesses:**

- There are concerns regarding the practical applicability of the proposed method. ABMS is computationally heavy as it requires M diffusion model operations at each step, and 1000 sampling steps.

- Most of the demonstrated tasks have limited practicality; for instance, the inpainting task only uses very small masks instead of large ones.

- The results are shown on pixel-space diffusion models. However, state-of-the-art diffusion models like Stable Diffusion or Flux-dev operate in latent space. Including results on these models would broaden the paper's scope.

**Questions:**

Is the proposed methodology applicable to recent state-of-the-art diffusion models such as Flux-dev?

---

> ### Author Response · Authors · 2025-11-22
>
> Thank you for your time reviewing our manuscript! Below, we will do our utmost to address each of your comments point by point:
> ***
>
> >### Weakness 1: Concerns about the time cost in practice and the sampling steps.
>
> Thank you for your question.
>
> - Taking image inverse problems as an example, we quantitatively evaluate the impact of our method on denoising step speed. When $M=1$ the time cost doubles compared with the original, as $M$ increases (provided the GPU memory limit is not exceeded), it grows proportionally.
>
>     | DPS/DSG   | ABMS M=1  | ABMS M=2  | ABMS M=3  | ABMS M=4  | ABMS M=5  |
>     |:--------: |:--------: |:--------: |:--------: |:--------: |:--------: |
>     | 4.8iter/s | 2.5iter/s | 2.1iter/s | 1.7iter/s | 1.5iter/s | 1.3iter/s |
>
> - Additionally, regarding the number of **sampling steps**, our method has been experimentally verified to be **compatible with various sampling and training strategies**, such as DDIM and flow matching. As described in Section 5.2, we employ 100-step DDIM sampling here. Even when guidance is applied at every iteration, the total generation time remains under one minute, which we believe is fully acceptable.
>
>
> >### Weakness 2: Concerns about the application scenery.
>
> - Firstly, inpainting, Gaussian deblurring, and super-resolution on ImageNet 256 are the most representative image inverse problems and have been widely adopted as a **standard benchmark** by related studies (e.g., [1, 2, 3, 4] so on).
>
> - More importantly, our approach has been validated across a broad range of application scenarios, data types, model architectures, and sampling strategies shown in the following table while demonstrating **consistent improvements**. We believe this sufficiently substantiates the generalizability of our method and its practical effectiveness in real-world applications.
>
>     | Data type         | Task                                  | Sample method            | Network             |
>     |:---------------- |:------------------------------------ |:----------------------- |:----------------- |
>     | Picture 256x256   | Linear inverse problem                | DDIM                     | Unet                |
>     | Picture 512x512   | Text style guidance                   | Flow matching to sde     | Stable Diffusion3.5 |
>     | Picture 256x256   | Faceid guided generation              | DDIM                     | Unet                |
>     | Trajectory        | Online Chinese Handwriting generation | DDPM                     | 1D CNN              |
>     | 3D graph          | Conditional molecule generation       | Euler-Maruyama method    | EGNN                |
>
>
> >###  Weakness 3 & question: If the method can be applied to latent diffusion and sota model?
> - **Pixel space is important**: We would like to emphasize that pixel space diffusion remains of **critical importance**. First, in domains such as molecular design, there is currently no universally acknowledged sota latent-space generative model. Second, for many conditional generation tasks (e.g., image inverse problems), the conditioning is formulated in the original space, making a piexel space generative model indispensable. Finally, even within the image domain, the most recent paper [6] continue to demonstrate the untapped potential of pixel space generative models.
>
> - **Can also apply to latent space**: To better alleviate this concern, we scaled our method to **Stable Diffusion 3.5** backbone in Section 5.4. Since this checkpoint was originally trained with the rectified flow objective, we first injected sampling-time stochasticity following [5] and then integrated our approach. Compared with the baseline, our method yields a **pronounced quality improvement**, producing specific stylized images. We believe this supplementary experiment further demonstrates the **scalability and practical effectiveness** of our method.
>
> In summary, our method is applicable to diffusion models operating in **both pixel and latent spaces**. It has demonstrated advantages not only at **small scales** but also within **large scale** architectures. We believe these empirical validations suffice to establish the method’s efficacy, universality, and significance, especially in application scenarios where guidance is crucial.
>
> [1] Diffusion Posterior Sampling for General Noisy Inverse Problems, ICLR 2023.
>
> [2] Manifold Preserving Guided Diffusion, ICLR 2024.
>
> [3] A Variational Perspective on Solving Inverse Problems with Diffusion Models, ICLR 2024.
>
> [4] Guidance with Spherical Gaussian Constraint for Conditional Diffusion, ICML2024.
>
> [5] Flow-GRPO: Training Flow Matching Models via Online RL, arXiv 2025.
>
> [6] Back to Basics: Let Denoising Generative Models Denoise, arxiv 2025.
>
>
> ***
> We appreciate your time and hope that our response can fully address your concerns, and that you will consider revising the rating.
> If there exits any further issues merit discussion, please do not hesitate to let us know!

---

> > ### Comment · Reviewer_Daim · 2025-11-27
> >
> > Thank you for your detailed rebuttal. My concerns regarding the time cost have been fully addressed. However, I still have reservations about the remaining two points.
> >
> > First, I agree that inpainting, Gaussian deblurring, and super-resolution on ImageNet 256 have long been considered standard benchmarks. When there were relatively few conditional diffusion models, it was indeed meaningful to control unconditional diffusion models at test time, and image restoration on ImageNet was a sufficiently challenging benchmark. However, in the current landscape, with numerous large-scale image restoration models [A, B], I am not fully convinced that these benchmarks remain equally informative nowadays. I would be very interested to hear the authors’ perspective on the continued significance of these tasks as primary benchmarks.
> >
> > Second, I also remain uncertain about the style transfer results in Section 5.4 using Stable Diffusion 3.5. From the provided samples, it appears that the structure of the corgi is largely preserved and only the color scheme is transferred. It is therefore unclear whether it is comparable to what is demonstrated in prior works such as StyleDrop [C] or RB-Modulation [D]. How about showing both qualitative and quantitative comparisons using the same style references as those prior works [C, D], so that the effectiveness of the proposed method on style transfer can be more clearly assessed?
> >
> > [A] Wu et al., One-Step Effective Diffusion Network for Real-World Image Super-Resolution, NeurIPS 2024.
> >
> > [B] Tan et al., OminiControl: Minimal and Universal Control for Diffusion Transformer, ICCV 2025.
> >
> > [C] Sohn et al., StyleDrop: Text-to-Image Synthesis of Any Style, NeurIPS 2023.
> >
> > [D] Rout et al., RB-Modulation: Training-Free Stylization using Reference-Based Modulation, ICLR 2025.

---

> ### Author Response · Authors · 2025-11-27
>
> Thank you for the clarification！ Nevertheless, we would like to re-emphasize our contributions and provide clarifications on your concerns:
> ***
> ### On the significance of training-free guidance：
> The core contribution of our work is a **general purpose**, training free guidance framework that outperforms prior methods. We do not target a single downstream task. As the reviewer correctly note *"in domains where conditional diffusion models are scarce, controlling an unconditional model becomes essential."*  While the CV community has recently developed specialized solutions for restoration or style transfer, **too many broader areas** e.g., handwriting sequence generation (§5.1) and molecular inverse design (§5.3) still **lack effective conditional diffusion** approaches. This is especially true when **the condition is a precise numerical vector** (e.g., molecular property), training a conditional generator is notoriously difficult (conditional EDM in our baseline for molecule design performs poorly, refer to Table 3). Our method can offer a **simple yet effective** remedy for these **under-served** domains, which I believe **holds considerable value and significance**.
>
> ### On the benchmarks:
> For the specific task of image restoration itself, dedicated conditional models already exist and work well. However, for evaluating the training-free guidance literature, every newly proposed method is expected to demonstrate gains on these standard benchmarks. We include diverse applications is precisely to **show that our method is consistently superior to existing guidance techniques**, not to claim state-of-the-art restoration results. More importantly, we appreciate the reviewer’s attention to the benchmark settings.  We fully **agree that benchmarks themselves must evolve** and indeed, this constitutes **a further contribution** of our paper:
> - We introduce the dual-focus evaluation protocol to better evaluate guidance quality.
>
> - We bring molecular inverse design into the guidance benchmark, which is a task that critically demands accurate guidance and has been absent from previous studies.
>
> ### On the SD3.5 based experiments:
>
>  In these experiments, our guidance objective is **simply** the distance between the Gram matrix of the CLIP image embedding and that of a style reference. Adding the corresponding gradient term during sampling with a proper step size preserves global structure while transferring color and texture into the reference style, actually demonstrating a clean **disentanglement of content and style** in the gradient.  The same phenomenon was also presented in paper [1], which adopted Stable Diffusion 1 as the backbone. To further show that our method can scale to larger models and flow-matching, we replaced the backbone with Stable Diffusion 3.5 while keeping all other settings unchanged.
>
>
> In addition, **given the positioning and aim of our method**, I believe our primary comparison target should be other training-free guidance methods, while the baseline guidance method yields noticeable artifacts, we believe this convincingly validates our approach.
>
>
> [1] Guidance with Spherical Gaussian Constraint for Conditional Diffusion, ICML2024.
>
> ***
> Based on the foregoing discussion, we fully understand that the reviewer, whose primary interest lies in concrete CV tasks, regards training-free guidance methods as having limited significance for that domain. We do not dispute this view, yet it it falls outside the scope of our core contribution.
>
> Correspondingly, the aspects of our principal contributions on which I believe we can reach **consensus with the reviewer** are in fact basicly aligned with the claims made in Section 1 of our manuscript:
>
> - 1. We introduce a dual-focus evaluation framework and new tasks to refine the benchmarking of guidance methods.
>
> - 2. We analyze the limitations of existing approaches and propose directions for improvement.
>
> - 3. The proposed method consistently outperforms prior training-free guidance baselines across all experimental settings, an outcome of critical importance for any discipline that relies on guidance based techniques.
>
>
>
> We sincerely hope that the reviewer will carefully reconsider our contributions in light of the **method's positioning**, the **values of training-free guidance field**, and its **current state of development**. While our method is not intended to *tackle vision tasks that have already been well addressed*, our goal is to *propose a better training-free guidance framework* and then can apply it to domains where **diffusion guidance is more urgently needed** in the future (e.g., molecule design, control, etc.). The vision tasks examined here are merely representative evaluation scenarios.
>
>
> Please let us know whether the above can address your concerns, if the reviewer still feel additional experiments under other settings are necessary, we will do our utmost to include them before the rebuttal deadline.
>
>
> Sincerely,
>
> The Authors.

---

> ### Author Response · Authors · 2025-11-27
>
> Dear Reviewer,
>
> Thank you again for your helpful remarks. In view of the tight rebuttal schedule, could you kindly indicate whether the supplementary  clarifications we provided alleviate your reservations? Any further advice would be most welcome and we are eager to incorporate it while time remains to strengthen the manuscript.
>
>
>
> Best regards,
>
> The Authors.

---

> > ### Comment · Reviewer_Daim · 2025-11-27
> >
> > According to the authors’ response, the proposed method works well not only on vision tasks but also in other domains where there are relatively few conditional diffusion models. Thank you for the prompt and detailed reply; all of my concerns have been addressed, and I raised my score accordingly.

---

> ### Author Response · Authors · 2025-11-27
>
> Thank you for your affirmation and positive feedback! We also appreciate our discussion period, which prompted us to further reflect on the positioning and significance of the method, enabling us to articulate its contributions more effectively.
>
> Sincerely,
>
> The Authors.

---

### Official Review · Reviewer_BtcN · 2025-10-29

**Soundness:** 3
**Presentation:** 2
**Contribution:** 2
**Rating:** 6
**Confidence:** 2

**Summary:**

This paper addresses a training-free guidance method to improve the inverse problem in diffusion models. The authors argue that the existing method, DPS, suffers from a biased gradient. To mitigate this, ABMS leverages the gradient of the averaged multiple backward predictions of diffusion models. With theoretical justification, ABMS shows improved results compared to existing baselines.

**Strengths:**

- **Well-Motivated and Simple Solution**: The proposed ABMS method is intuitive, well-motivated by the law of total expectation, and directly targets the identified source of bias (single-point estimation). It seems to be easily adapted to the existing codebase due to its simplicity and can be widely utilized as it does not require any additional conditions.

- **Comprehensive Experiments**: The method's effectiveness is demonstrated across three different domains. The consistent improvements across all tasks support the claims.

- **Theoretical Analysis**: The estimation error analysis showing that ABMS's error bound under the given assumptions avoids the $\delta_{f}(x_t)$, which plagues the DPS bound, provides a theoretical justification for the effectiveness of the ABMS.

**Weaknesses:**

- **Computational Overhead**: The most significant weakness is the increased computational cost. ABMS requires $M$ denoising network evaluations in addition to the original denoising steps. While it can be parallelized, the memory consumption can grow rapidly as it also requires additional gradient calculations. For a more comprehensive analysis, the additional computational time and memory consumption for ABMS should be reported.

- **Novelty in Context**: The idea of using Monte Carlo sampling to get better estimates in diffusion guidance is not entirely new (e.g., LGD-MC). The paper's novelty lies in the one-step-back formulation. The paper could be stronger if it more directly compared against a simpler MC-DPS baseline (i.e., averaging $M$ estimates from $x_t$, not $x_{t-1}$) to isolate the benefit of the backward step from the benefit of MC sampling. Furthermore, while the authors argue that LGD-MC incurs high computational costs, the proposed ABMS is a more computationally intensive method as it requires multiple diffusion model calls with gradient calculation, while LGD-MC requires only one diffusion model call.

- **Limited Scope of Samplers**: While the authors acknowledge this limitation, it is unclear how ABMS, which relies on the SDE-based one-step transition, would be adapted to faster ODE-based or higher-order solvers (like DPM-Solver++, etc.) that are now state-of-the-art for fast sampling. This potentially limits the method's practical application.

**Questions:**

- **Clarification on Computational Cost**: Please refer to Weakness 1.

- **Ablation Study Without DSG**: ABMS opts for the DSG for the stability of the algorithm. It could be helpful if the base performance of ABMS without DSG is analyzed compared to DPS.

**Details Of Ethics Concerns:**

No ethical concerns are raised.

---

> ### Author Response · Authors · 2025-11-22
>
> We sincerely appreciate your recognition of our motivation and experimental validity, as well as the constructive suggestions you provided! Below, we will address each of your concerns point by point.
> ***
>
> >### Weakness 1& question 1:  About the computional overhead analysis.
>
> - Tanks for your suggestion, we have supplemented the analysis on how the time cost scales with the number of sampling steps and add it in the Appendix A.2. Using the image inverse problem as an example, the following table reports the denoising iterations per second under different values of $M$. When $M=1$ the time cost doubles compared with the original. As $M$ increases, it grows proportionally. Since our method has been validated to be compatible with efficient samplers (answer to weakness 3), the computational overhead is **fully acceptable** in practical applications.
>
>
>     | DPS/DSG   | Ours M=1  | Ours M=2  | Ours M=3  | Ours M=4  | Ours M=5  |
>     |:--------: |:--------: |:--------: |:--------: |:--------: |:--------: |
>     | 4.8iter/s | 2.5iter/s | 2.1iter/s | 1.7iter/s | 1.5iter/s | 1.3iter/s |
>
> - As for memory overhead, it varies with datasets, batch size and model architectures. Using the molecule inverse design problem as an example, the batch size is set as 50:
>
>     | DPS/DSG   | Ours M=1  | Ours M=2  | Ours M=3  | Ours M=4  | Ours M=5  |
>     |:--------: |:--------: |:--------: |:--------: |:--------: |:--------: |
>     | 5094MB | 8412MB | 16573MB | 24436MB | 32889MB | 41501MB |
>
>     In most tasks we set $M$ = 3 or 5 because these values can provide parallelism as well as satisfying performance.
>
>
> >### Weakness 2:  About the comparison with LGD-MC.
>
> Thank you for raising this point! After careful verification, we confirm that under the same number of sampling steps, our method does incur a larger computational cost. We have therefore revised the original claim and now provide an updated comparison with LGD-MC:
>
> - Intuitively, LGD-MC assumes that $p(x_0 | x_t) $ is a Gaussian distribution and directly injects noise into the single point estimate of $x_0$. Because the true conditional distribution is highly complex, this assumption is intrinsically limiting: even an infinite number of samples cannot capture its multi modal nature. In contrast, our approach first draws several stochastic intermediate states from $x_t$, then estimates $x_0$ from each of these states. This procedure leverages the non-linearity of the network and naturally accommodates multi modal distributions.
>
> - We have also additionally included an ablation on the writing task (Section 5.1) in which all hyper parameters: step size and guidance scale fixed at 0.1 are identical, only the guidance direction differs. With M = 3, our method already yields a clear improvement, while further increasing M to 5 or 10 brings only marginal gains.
>
>     | Method       | DSG   | LGD (M=5) | LGD (M=10) | Ours (M=3) |
>     |:-----------: |:----: |:-----------: |:------------: |:---------: |
>     | Style Score↑ | 0.534 |     0.553     |     0.571      |    0.878    |

---

> ### Author Response · Authors · 2025-11-25
>
> >### Weakness 3: If our method can apply to other samplers.
>
> - Although our method was originally motivated by single step DDPM, we find in practice that it remains **effective** and **consistently outperforms** the baseline even when higher order solvers are used. In general, the key idea is to inject a stochastic intermediate state $x_s$ between $x_t$ and $x_0$, without requiring $s$ to be exactly $t−1$. For example, in the image inverse problems (Section 5.2) we use 100 step DDIM, and in the supplementary experiments (Section 5.4, Figure 4) we show that the benefits still hold (**even more significant**) when applying to Stable Diffusion 3.5 trained with rectified flow. In summary, we have valided our method across the following settings:
>
>      | Data type         | Task                                  | Sample method            | Network             |
>      |:---------------- |:------------------------------------ |:----------------------- |:----------------- |
>      | Picture 256x256   | Linear inverse problem                | DDIM                     | Unet                |
>      | Picture 512x512   | Text style guidance                   | Flow matching to sde     | Stable Diffusion3.5 |
>      | Picture 256x256   | Faceid guided generation              | DDIM                     | Unet                |
>      | Trajectory        | Online Chinese Handwriting generation | DDPM                     | 1D CNN              |
>      | 3D graph          | Conditional molecule generation       | Euler-Maruyama method    | EGNN                |
>
> - The few-step generation methods referred to in our limitations section pertain to recently proposed generative approaches designed to accomplish sampling in a single step or with an extremely limited number of steps (e.g., 4, 8), such as Consistency Models and MeanFlow. Whether guidance methods can be effectively applied to these novel approaches remains an open question.
>
> >### Question2: The ablation without DSG.
>
> Thank you for this insightful question. Below we carefully clarify the distinctions among DPS, DSG and our approach. Conceptually, guidance proceeds in two stages: (1) determining the descent direction and (2) choosing the step size. DPS provides the most basic gradient direction, while the step size is completely controlled by a hyper parameter. DSG refines this by computing the largest possible step that still keeps the update on the data manifold, but the direction itself remains identical to DPS. Hence DSG can be viewed as an augmented version of DPS. Our contribution goes one step further: we optimize the gradient direction.
>
> - Consequently, we regard **DSG itself** as an ablated baseline of our framework.
> - To better address this concern, we removed the step size trick of DSG for linear inverse problems on images and used the same hyper parameters as the original DPS. We can still observe **consistent performance improvements**.
>
>     | Task               | Method | PSNR↑  | SSIM↑ | LPIPS↓ | FID↓  |
>     |--------------------|--------|--------|--------|--------|--------|
>     | Inpainting         | DPS    | 27.56  | 0.825  | 0.251  | 30.57  |
>     |                    | Ours   | **28.13** | **0.847** | **0.214** | **25.25** |
>     | Gaussian Deblur    | DPS    | 18.78  | 0.387  | 0.523  | 52.13  |
>     |                    | Ours   | **21.03** | **0.471** | **0.455** | **46.75** |
>     | Super Resolution   | DPS    | 22.07  | 0.633  | 0.317  | 41.36  |
>     |                    | Ours   | **23.01** | **0.665** | **0.295** | **37.23** |
>
>
> ***
> We hope that our reply can satisfactorily resolve your concerns and do not hesitate to inform us if anything else arise!

---

> > ### Comment · Reviewer_BtcN · 2025-11-27
> >
> > After reading the author's feedback, I am still confused about how the proposed method can be applied to a multi-step sampler like DPM-Solver. However, I think this is a minor one because recent faster diffusion models adopt the flow matching strategy. Given that, I keep my positive rating.

---

> ### Author Response · Authors · 2025-11-27
>
> Thank you for your prompt and constructive response. You are absolutely right: because DPM-Solver is a *purely deterministic* ODE solver with *no stochastic extensions*, it is inherently incompatible with our method.  We have added the corresponding references and an extended discussion of this point in the revised manuscript (Appendix A.2).
>
> As you also noted, DDIM as well as more recent flow matching based methods, though originally ODE driven, have already been equipped with stochastic variants in the literature. We have therefore included these algorithms in our experiments to demonstrate the broader applicability of our approach. We also believe it is enough to make the computational overhead fully acceptable in practical.
>
> Once again, thank you for your insightful suggestion and for your positive rating!

---

### Official Review · Reviewer_93q3 · 2025-11-02

**Soundness:** 2
**Presentation:** 3
**Contribution:** 3
**Rating:** 4
**Confidence:** 4

**Summary:**

The paper targets bias in training-free diffusion posterior sampling (DPS) for conditional generation and inverse problems. DPS approximates the posterior p(x0∣y)∝p(x0)exp⁡[−L(x0;y)]p(x_0|y) through two linearizations: (1) moving the expectation inside the loss, and (2) replacing the conditional mean with the denoiser output via Tweedie’s formula. The authors argue these lead to biased gradients and propose ABMS (Additional Backward step with Monte-Carlo Sampling): before estimating guidance at step ttt, it samples xt−1∼ p(xt−1∣xt) multiple times, evaluates the loss on each denoised x^0(xt−1), and averages the results to reduce bias. A dual-focus evaluation (alignment vs. global quality) is introduced. Experiments on classifier-guided digit synthesis, image inverse problems (SR, inpainting, deblurring), and molecular property conditioning show modest but consistent gains.

**Strengths:**

The problem—bias in loss-guided diffusion—is relevant and well-motivated.

The proposed method (MC sampling one step earlier) is simple, plug-and-play, and compatible with existing samplers. Also parallelization of MC sampler justifies the added computations.

Evaluation across multiple domains, with clear quantitative metrics.

**Weaknesses:**

The “unbiased” claim is overstated: ABMS still produces a lower-bias approximation but not a provably unbiased gradient of the tilted posterior. No formal unbiasedness proof or convergence result is provided.

The theoretical bound (Sec. 4.2) only compares error upper bounds under strong assumptions (Lipschitz fff, monotone denoiser accuracy). Variance of the stochastic gradient is unaddressed.

Scope limited to DPS. Extensions to other plug-and-play or variational samplers (e.g., RED-Diff, flow-matching, ODE solvers) are not discussed experimentally.

The empirical improvements, though consistent, are incremental; figures often lack statistical significance or ablations isolating the MC vs. scaling effects.

**Questions:**

Can the authors clarify whether the proposed MC sampling yields an unbiased estimator of gradients? If not, what assumptions make the bias negligible?

Does the ABMS correction still hold under deterministic DDIM/flow samplers?

How does the method compare to re-scoring or re-noise-based variance-reduction schemes such as RED-Diff or Score-DPO?

Are there ablations showing the effect of the hypersphere projection alone versus MC averaging alone?

For molecular tasks, how sensitive is performance to the sampling count MMM?

---

> ### Author Response · Authors · 2025-11-22
>
> Thank you for the time you devoted to reviewing our manuscript and for affirming the strengths of our method as well as for your constructive suggestions! In the following, we will respond to each of your concerns point by point.
> ***
> >### Weakness 1, 2 & Question 1: If the proposed method is unbiased as well as issues with theoretical proof.
>
> - **Unbiased part**: We appreciate your attention to our methodology. Indeed, our approach does not guarantee unbiasedness, nor do we claim it in our paper. Since $p(x_0|x_t)$ is almost as complex as the data distribution and essentially **intractable**, while diffusion models can only produce a single point estimate of $x_0$, to the best of our knowledge, no existing method for estimating $E[f(x_0)|x_t]$ can provide unbiasedness guarantees.
>
> - **Theoretical part**:  Similarly, due to the complexity of the data distribution, the arbitrariness of the conditional function $f$, and errors in the generator's estimation of $x_0$, directly proving gradient reliability *at every single point seems impossible*. Existing methods such as DPS[1], MPGD[2], and DSG[3] have indeed acknowledged the Jensen gap problem and also prove its *upper and lower bounds under some strong assumptions* (e.g the Lipschitz  f), but do not adopt strategies to directly reduce the impact of the Jensen gap. We have also further revised the theoretical proof in the paper.  As shown in Proposition 1, we actually only establish that the expected estimation error upper bound of our method for the true $E[f(x_0)|x_t]$ is lower than that of previous methods.
>
> - More importantly, we believe that truly **essential** and **convincing** validation lies in the **consistent improvements** across diverse scenarios and showing **particularly significant improvements** in some certain cases. In this context, our proof aims to *account for the likely sources of the gains* offered by our method (the reconstruction term and the Jensen gap term) based on as few empirically well accepted assumptions as possible, while also *pointing out directions for future improvement*.
>
> >### Weakness 3 & Question 2,3:  A.  If our method can adapt to ODE sampler, even flow matching?  B. Compare with variational samplers, e.g Red-diff.
>
> - **A. Apply to DDIM and Flow matching:**
> Our method is not intrinsically tied to DDPM and it can be coupled with any stochastic sampler. For example, in the image inverse-problem discussed in Section 5.2 we used a 100 step **DDIM**. Furthermore, in the supplementary experiment (Section 5.4) we followed [4] to enable SDE sampling of Stable Diffusion 3.5（based on **rectified flow**）and observed **substantial quality gains over the baseline (Figure 4)**. These results confirm that the advantage of our approach generalizes across different **sampling strategies**, **training strategies** and even **larger scale architectures**.
>
> - **B. Discussion with Missing Baseline, e.g. Red-Diff:**
> Thank you for this valuable suggestion. We have added the missing citation (in the introduction part and the related work) and faithfully implemented Red-Diff [5] under exactly the original hyperparameters and settings for the inpainting task (e.g. Adam optimizer with1000steps, $\lambda=0.25$, $lr=0.5$,  and so on). As shown in the table below, our approach delivers slightly better quantitative metric.
>
>     Moreover, the original Red-Diff paper focuses on image domain tasks, we actually find that it can also in principle handle conditional generation tasks given any differentiable target, in a different way from guidance based methods. We have *restated this observation in the related work section* to provide a more complete research background. It solves the problems by optimization, uses the condition as a loss function with diffusion prior as a regularized loss. The computational overhead per iteration for it is minor, yet it demands numerous optimization steps (e.g., 1000) for efficacy. In contrast, our method is a guidance technique for traditional diffusion generation. Our method's denoising steps are more computationally intensive, but can be utilized with higher order samplers (e.g., 50-100 DDIM steps). Thanks again for your valuable addition!
>     | Method   | PSNR↑  | SSIM↑  | LPIPS↓  | FID↓   |
>     |:-------: |:-----: |:-----: |:------: |:-----: |
>     | DSG      | 28.67 | 0.877 | 0.176  | 23.63  |
>     | Red-Diff | 28.76  | 0.871  | 0.1836  | 19.95  |
>     | Ours     | **29.23** | **0.889** | **0.154**  | **19.25**  |
>
>
> ***
>
> [1] Diffusion Posterior Sampling for General Noisy Inverse Problems, ICLR 2023.
>
> [2] Manifold Preserving Guided Diffusion, ICLR 2024.
>
> [3] Guidance with Spherical Gaussian Constraint for Conditional Diffusion, ICML2024.
>
> [4] Flow-GRPO: Training Flow Matching Models via Online RL, arXiv 2025.
>
> [5] A Variational Perspective on Solving Inverse Problems with Diffusion Models, ICLR 2024.

---

> ### Author Response · Authors · 2025-11-22
>
> >### Weakness 4 & Question 4: About the improvement statistic significance and ablation study.
>
> Thank you for raising this important point.
>
> - **Regarding the consistent improvement**.  To ensure full reproducibility, every experiment was run under a fixed random seed and quantitative metrics are reported as the mean of three independent runs. Therefore, all results are **statistically significant**. Moreover, a key contribution of our work is the consistent improvements across **diverse scenarios** presented in the following Table, which sufficiently demonstrates the **generalizability and effectiveness** of our method.
>
>     | Data type         | Task                                  | Sample method            | Network             |
>     |:---------------- |:------------------------------------ |:----------------------- |:----------------- |
>     | Picture 256x256   | Linear inverse problem                | DDIM                     | Unet                |
>     | Picture 512x512   | Text style guidance                   | Flow matching to sde     | Stable Diffusion3.5 |
>     | Picture 256x256   | Faceid guided generation              | DDIM                     | Unet                |
>     | Trajectory        | Online Chinese Handwriting generation | DDPM                     | 1D CNN              |
>     | 3D graph          | Conditional molecule generation       | Euler-Maruyama method    | EGNN                |
>
> - **Regarding the magnitude of gain**: While delivering consistent improvements across all scenarios, we would like to emphasize that **on harder and more non-linear** problems, it seems our method can yield **larger benefits**. For example, in tasks of handwriting generation **(Section 5.1: style score from 0.53 to 0.88)** and SD3.5 style guidance **(Section 5.4: from images with artifacts to clean ones)**, our method both show **dramatic improvements** over the respective baselines. We believe this sufficiently demonstrates the method’s effective contribution to existing work.
>
> - **Regarding the ablation**: The guidance process actually consists of two steps: one is direction finding, and the other is step size determination. Our ABMS aims at identifying a better guidance direction based on plain DPS, while the projection in DSG seeks the maximum step size that preserves the data manifold. Our contribution lies in direction finding while keeping the other settings same as DSG. Therefore, **DSG itself** actually constitutes ablation of our MC vs projection.  Moreover, to better address this concern, we also removed the step size trick of DSG for linear inverse problems on images and used the same hyper parameters as the original DPS. We can still observe the **consistent performance improvements**.
>
>     | Task               | Method | PSNR↑  | SSIM↑ | LPIPS↓ | FID↓  |
>     |--------------------|--------|--------|--------|--------|--------|
>     | Inpainting         | DPS    | 27.56  | 0.825  | 0.251  | 30.57  |
>     |                    | Ours   | **28.13** | **0.847** | **0.214** | **25.25** |
>     | Gaussian Deblur    | DPS    | 18.78  | 0.387  | 0.523  | 52.13  |
>     |                    | Ours   | **21.03** | **0.471** | **0.455** | **46.75** |
>     | Super Resolution   | DPS    | 22.07  | 0.633  | 0.317  | 41.36  |
>     |                    | Ours   | **23.01** | **0.665** | **0.295** | **37.23** |
>
> >### Question 5 : How sensitive is the performance to sample num in molecular task
>
> We report the performance of our method on the molecular task (mu) for varying numbers of sampling steps. The performance gain increases substantially from M = 1 to M = 3, but slows down as M grows further. Similar observations also hold for image inverse problem tasks. We therefore set M = 3 to balance efficacy and computational cost in this task.
>
> | Method | DSG    | M=1    | M=3    | M=5    |
> |:-----: |:-----: |:-----: |:-----: |:-----: |
> | MAE↓   | 0.7811 | 0.7652 | 0.7274 | 0.7042 |
> | MS↑    | 0.8001 | 0.7986 | 0.8059 | 0.7927 |
>
> ***
> We sincerely appreciate your insightful feedback which has strengthened the completeness of our paper. We hope that our reply can satisfactorily resolve your concerns!

---

> ### Author Response · Authors · 2025-11-27
>
> Dear Reviewer,
>
> We are grateful for your insightful feedback. With only a few days remaining for the rebuttal, we would appreciate knowing whether our newly supplied experiments and explanations satisfactorily resolve your concerns. Should you have any additional guidance, kindly inform us at your earliest convenience so that we can allocate the remaining time to refine the manuscript accordingly.
>
>
> Best regards,
>
> The Authors.

---

### Official Review · Reviewer_vUip · 2025-11-11

**Soundness:** 2
**Presentation:** 3
**Contribution:** 2
**Rating:** 4
**Confidence:** 4

**Summary:**

This paper addresses a critical and widely recognized problem in training-free guided diffusion models: the degradation of sample quality (e.g., FID) under strong conditional guidance. The authors convincingly argue that this issue stems from a systematic estimation error and bias in the guidance gradient, which is typically derived from a single, noisy denoising step. To mitigate this, they propose ABMS, a plug-and-play Monte-Carlo sampling strategy. At each reverse diffusion step, instead of relying on a single estimation path, ABMS explores multiple (M) potential predecessor states, denoises each one to predict a clean output, and then averages the guidance function evaluations across these M predictions. This Monte-Carlo averaging yields a more stable and accurate estimate of the true guidance gradient. The paper provides a theoretical justification (Proposition 1) showing that ABMS achieves a lower estimation error bound compared to the standard DPS approach. The effectiveness of ABMS is demonstrated empirically across a diverse set of tasks, including stylized handwriting generation, standard image inverse problems (inpainting, super-resolution, deblurring) on ImageNet, and molecular property prediction. The results consistently show that ABMS allows for stronger guidance without the typical collapse in sample quality, outperforming existing methods.

**Strengths:**

1. The paper tackles a fundamental challenge in guided diffusion. The "dual-focus evaluation" paradigm, which explicitly calls for balancing task-specific performance (e.g., reconstruction error) with general sample quality (e.g., FID), is an excellent and necessary framing for this problem area. The authors clearly articulate why strong guidance often leads to poor results, providing strong motivation for their work.
2. The proposed ABMS method is simple, well-motivated, and elegant. The core idea of using Monte-Carlo averaging to obtain a more robust estimate of an expectation is a classic statistical principle applied very effectively here. Its "plug-and-play" nature makes it broadly applicable to various diffusion frameworks and tasks without requiring model retraining, which is a significant practical advantage.
2.1 While not a deep theoretical paper, the inclusion of the estimation error analysis and Proposition 1 provides a solid theoretical grounding for why ABMS should be expected to work better than single-path estimators. It connects the intuitive idea of averaging to the mathematical problem of reducing the bias introduced by Jensen's inequality.

3. The experimental section is a major strength of this paper.
  - Diversity of Tasks: Testing the method on stylized text, natural images, and molecular data convincingly demonstrates its generality.
  - Rigorous Evaluation: The use of performance curves (Figure 3) that plot task-specific distance against FID is particularly effective. These plots provide a clear and powerful visualization of the core contribution, showing that ABMS dominates other methods by achieving a better trade-off frontier.

**Weaknesses:**

1. The primary drawback of ABMS is the increased computational cost, which scales with the number of Monte-Carlo samples, `M`. The paper demonstrates the effectiveness of `M=3` and `M=5` but never explicitly analyzes or reports the trade-off between performance and inference time/FLOPs. For a sampling method, this performance-cost analysis is crucial for researchers and practitioners to assess its viability. While Figure 3 implicitly shows the performance gain for different `M`, the associated cost is not quantified.
2. While the use of ImageNet 256x256 is a standard and respectable benchmark, the field of generative models is rapidly moving towards much higher resolutions (512x512, 1024x1024) and significantly larger models (e.g., Stable Diffusion). The paper does not demonstrate whether the benefits of ABMS hold or are even more critical at this larger scale, where guidance is often essential. Demonstrating scalability to at least one high-resolution setting would substantially increase the paper's impact.
3. The paper would benefit from a more detailed discussion of how ABMS relates to other methods that also aim to improve guided sampling by investing more computation. A key missing comparison is with "Restart Sampling"[1]. Both methods address quality degradation under strong guidance but seem to operate on different principles. A discussion clarifying these differences would better situate the paper's contribution within the current literature.

Refrence:
[1] Xu, Yilun, et al. "Restart sampling for improving generative processes." Advances in Neural Information Processing Systems 36 (2023): 76806-76838.

**Questions:**

See the weakness.
While the Weakness.3 is the most important, which affects the innovation of this paper, with an excellent explanation, I will increase the score, while a bad explanation, I will decrease the score.

---

> ### Author Response · Authors · 2025-11-22
>
> We greatly appreciate the reviewer’s recognition of the strengths of our method and their constructive feedback！In what follows, we provide a detailed response to each of your suggestions and questions.
> ***
> > ### Weakness 1: Lacking of the analysis of the inference time cost for different $M$.
>
> - We have supplemented the results on how time cost varies with the number of sampling steps in Appendix A.2, which we believe makes our paper more complete. Taking the image inverse problem as an example, the table below shows the number of denoising iterations per second for the diffusion model under different values of $M$.
>
>     | DPS/DSG   | Ours M=1  | Ours M=2  | Ours M=3  | Ours M=4  | Ours M=5  |
>     |:--------: |:--------: |:--------: |:--------: |:--------: |:--------: |
>     | 4.8iter/s | 2.5iter/s | 2.1iter/s | 1.7iter/s | 1.5iter/s | 1.3iter/s |
>
> - As can be seen, when $M=1$ the time cost doubles compared with the original, as $M$ increases (provided the GPU memory limit is not exceeded), it grows proportionally. While we employ a 100-step DDIM sampler, even when guidance is applied at every iteration, the complete sampling run takes less than a minute. Given that our method has been validated to be compatible with efficient samplers, we believe the computational overhead is **fully acceptable** in practical applications.
>
>
> >### Weakness 2: If the method can apply to larger model, higher resolution picture.
>
> Thank you very much for this constructive suggestion! To address the concern, we conducted an in-depth investigation and scaled our method to a larger model and higher resolution generation. Specifically, we adopted **Stable Diffusion 3.5** to produce **512 × 512** pixel images. Because this checkpoint was trained with the flow matching objective (rectified flow), we introduced stochasticity at sampling time by following [1] and then incorporated our approach.
>
> The qualitative results are provided in section 5.4 in the revision. Compared with the baseline (DSG), our method yields **noticeably clearer** and **higher quality** stylized images. This additional experiment demonstrates that our approach can scale to substantially larger models and remains still effective for models trained with flow matching objectives. We believe these findings substantiate the scalability of our method.
>
> >### Weakness3: Missing a discussion with a relative method.
>
> Thank you for prompting us to strengthen the research background. After carefully reading [2] and several related works, we have added a new paragraph to Section 2 (Related Work) that places our contribution in a broader context. For the key missing method:
>
> - **Commonality**：Both [2] and our method trade additional computation for higher sample quality.
>
> - **Difference**：Paper [2] iteratively reinjects a carefully scaled noise during the generative process and then redenoises, eliminates accumulated errors along the denoising path, thereby squeezing more out of the **internal** denoising capacity of the pretrained model and it effectively improves the quality of sampling. Our method, in contrast, assumes an arbitrary differentiable external condition and spends the extra compute to obtain a more accurate **external** guidance gradient for conditional sampling.
>
> - The two techniques seem orthogonal and can be used together: one can apply the renoising denoising cycle with our guidance step at the same time if necessary.
>
> In addition, there are other works that are mainly based on noise search (refer to Section 2).  All of these methods provide effective options for increasing computation and improving sample quality. Compared to these related approaches, a distinctive feature of our method is that by obtaining a more accurate guidance gradient, it becomes **more well suited** for inverse design tasks that impose **precise numerical requirements**.
>
> [1] Flow-GRPO: Training Flow Matching Models via Online RL, arXiv 2025.
>
> [2] Restart Sampling for Improving Generative Processes, NeurIPS 2023.
>
> ***
> Thank you once again for your valuable suggestions, which have helped us make the manuscript more complete! We hope our response has fully addressed your concern. If any further issues remain, please feel free to let us know.

---

> ### Author Response · Authors · 2025-11-27
>
> Dear Reviewer,
>
> Thank you very much for your constructive comments. Given that the rebuttal period spans only a few days, we sincerely wonder whether the additional experiments and clarifications we have provided effectively address your concerns. If you have any further suggestions, please do not hesitate to let us know so that we still have time to improve our manuscript to the greatest extent possible.
>
> Best regards,
>
> The Authors.

---

### Author Response · Authors · 2025-11-23
**General Response**

We thank the reviewers for their thorough and constructive comments and AC's organization in reviewing our paper. We are glad that the reviewers recoginze our method as **well-motivated** (vUip, 93q3,  BtcN) and supported by **solid experimental validation**(vUip, 93q3, BtcN, Daim) **across 5 diffenrent benchmark**.
***

Based on the reviewers' valuable feedback, we have performed additional experiments and updated the manuscript, which resolves the reviewers’ concerns. The major additional experiments and improvements are as follows:

1. In Section 5.4, we have applied our method to the sota latent diffusion model Stable Diffusion 3.5, which is based on rectified flow according to the suggestion of reviewers  vUip, 93q3, BtcN and Daim. We believe this supplementary experiment further validates our method and **broadens its scope of application**. We want to highlight that our method achieves consistent performance gains over the baseline across all the following scenarios, with **notable improvements** in certain cases.

     | Data type         | Task                                  | Sample method            | Network             |
     |:---------------- |:------------------------------------ |:----------------------- |:----------------- |
     | Picture 256x256   | Linear inverse problem                | DDIM                     | Unet                |
     | Picture 512x512   | Text style guidance                   | Flow matching to sde     | Stable Diffusion3.5 |
     | Picture 256x256   | Faceid guided generation              | DDIM                     | Unet                |
     | Trajectory        | Online Chinese Handwriting generation | DDPM                     | 1D CNN              |
     | 3D graph          | Conditional molecule generation       | Euler-Maruyama method    | EGNN                |


2. In Appendix A.2, we have provided the analysis of the computational overhead of our method according to the suggestion of reviewers  vUip, 93q3, Daim.  As our method is applicable to high order samplers, its overhead is shown to be completely acceptable for real world use.

3. We have thoroughly investigated and analyzed related methods, including **Restart Sampling** (reply to reviewer vUip), **Red-Diff** (reply to reviewer 93q3), and **LGD-MC** (reply to reviewer BtcN). We have supplemented the required **comparative experiments**, carefully elaborated on the connections and differences between our method and these related works, and thereby situated the paper’s contributions within a broader scope.

4. We have supplemented the relevant ablation studies to make the effectiveness of the method more convincing.

5. We have also refined the proofs in the theoretical section, making it more rigorous and providing explanations for the performance gains achieved by the method.

***
The above revisions have been integrated into our final manuscript. We have tried our best to address each reviewer’s specific concerns separately, and our responses are included beneath each review for reference. We sincerely appreciate the reviewers’ valuable suggestions!

---

### Meta-Review · Area_Chair_Zrp7 · 2026-01-06

**Summary:**

The manuscript proposes additional backward denoising and Monte Carlo sampling steps to achieve better performance in guidance models. The approach is validated by empirical performance.

**Reviewer Concerns:**

The authors have addressed most concerns raised by the reviewers.

**Reviewer Scores:**

I believe the reviewers would likely increase their scores towards acceptance.

---

### Decision · Program_Chairs · 2026-01-26

Accept (Poster)